# Unpacking the Principal Strategies in Leveraging Weighted Student Funding

Chun Sing Maxwell Ho 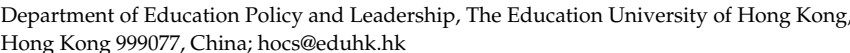

Department of Education Policy and Leadership, The Education University of Hong Kong, Hong Kong 999077, China; hocs@eduhk.hk

**Abstract:** Weighted student funding (WSF) systems have been implemented in various countries to give schools more autonomy over how to allocate their funding. School principals use funding to maintain school operations and foster innovation for achieving educational goals. However, despite the importance of this process, scholarly research has largely overlooked how principals make decisions about allocating their financial resources. Accordingly, this study seeks to provide practical insights into the strategies used by one school by highlighting their staff's perceptions about using their WSF to maintain school operations and spur innovation. Using a case study approach, we investigated a principal who effectively used a school's WSF to transform a failing school into an innovative one. The findings revealed that the principal strategically implemented financial management mechanisms in a way that inspired teachers to consider more profoundly how a school's WSF can help achieve educational goals. The principal fostered consensus on the school's direction, encouraged innovation through hands-on experiential learning and strategic planning, and facilitated funding for innovative teachers by guiding proposal development. In the final section of this article, insights into the shifting cultural and practical landscape of financial resource utilization within schools are discussed.

**Keywords:** weighted student funding; school financial management; principal leadership; school innovations; failing school

## 1. Introduction

The function of schools is to enhance teaching efficiency to improve student learning outcomes while meeting the evolving needs of society [1–3]. By introducing innovation in teaching and learning, schools can improve student outcomes, leading to better-prepared graduates while positively impacting society [4]. Nonetheless, spurring innovation requires adequate funding, which is often lacking in weighted student funding (WSF). In most countries, schools primarily rely on their WSF to maintain school operations and promote innovation, while generally being required to use all allocated funds within the school year [5]. Despite significant financial investments by governments, WSF is often insufficient. High staffing costs burden schools as they must maintain a certain teacher-to-student ratio, leaving little money from WSF for small-scale initiatives [6]. This has restricted the ongoing enhancement of schools. Therefore, in the case of WSF, effective school financial resource management is critical to allocate limited resources from public budgets efficiently to support the sustainability of school innovations [7]. Over the past two decades, there has been a growing recognition of the important role played by school principals in ensuring the effective use of WSF [8,9]. Despite this recognition, few studies, if any, have explored the efficacy of the specific strategies employed by principals to leverage WSF to benefit school operations and increase innovation [10–12].

Promoting sustainable innovation is essential to enhance teaching and learning and improve student outcomes, and effective school financial resource management is critical to support school operations and education innovation. The primary objective of the

present study is to provide new insights into how a principal enabled successful school operations and spurred innovation, specifically through the efficient management of the school's WSF. Accordingly, the study is guided by three research questions: (1) What is the principal's perception of utilizing the school's WSF? (2) What strategies does the principal employ to manage WSF for maintaining school operations and promoting innovation? Notably, school financial decision-making is often a collaborative effort involving various stakeholders, including principals and teachers [13]. Therefore, a third question was devised: (3) How do teachers react to a principal's strategies in managing the school's WSF? By using this approach, this study provides greater insight into using WSF to realize sustainable innovation and educational objectives.

## 2. Literature Review

### 2.1. Weighted Student Funding (WSF)

WSF is a school funding model that allocates funds based on the individual needs of each student. A fixed amount of money from the government is allocated for each student, with additional funds for those requiring extra support, including students with special needs, second-language learners, or those from low-income families [5]. All unspent school funds are returned to the government treasury (call-back), although most schools tend to use the funds carefully at the beginning of the year and spend any remaining money at the end of the year [14]. The WSF system is prevalent in countries with decentralized education systems, such as the United States and the Netherlands, which grant schools more autonomy [15,16]. Although the WSF mechanism can vary, depending on the country, a shared objective of allocating financing based on student needs is typically employed [15,17].

WSF plays a pivotal role in ensuring the sustainability of school operations and fostering continuous improvement. Its implementation is primarily directed toward two significant areas: sustaining daily school operations and fostering innovation. Schools utilize their WSF to distribute resources across various day-to-day operations, including but not limited to staff salaries, materials, technology, professional development, and other programs and services [6,18]. Schools also use their WSF to encourage innovation and enhance their teaching and learning approaches in response to the evolving educational environment [15,19]. However, for schools to successfully facilitate innovative teaching and learning practices, additional funding beyond WSF must be sought for teacher training, staff relief, and upgrading facilities, among others [15]. Thus, to secure funding for innovative practices, schools often submit proposals to donors or government agencies that are aligned with the nature of the innovation in question [19,20]. This type of funding, however, lacks flexibility, particularly when it comes to the granularity of each expenditure, which often forces schools to craft initiatives that match the criteria and preferences set by the funding source, rather than meet the needs of their students [20]. Given the constraints of securing extra funding, schools must monitor their budgets effectively to support and sustain teaching and learning initiatives [21].

### 2.2. The WSF Mechanism within Schools

In reviews of the literature on financial resource management, McKinney and Yizengaw & Agegnehu [22,23] suggest that managing financial resource mechanisms in schools involves five iterative steps that school principals should use to sustain innovations that align with their goals.

First, budget preparation requires the establishment of well-defined goals and desired outcomes, particularly concerning advocating initiatives [22]. To achieve this requirement, the goals must be identified and the necessary funding provided to attain them, leading to the development of a long-term investment plan for deploying resources. School leaders should allocate funding according to their needs, aligning with their overall desired outcomes [24,25].

Second, setting objectives for operational units is an essential component of budget preparation [22]. Within this process, each department and working committee should

identify specific goals that align with the school's overall objectives [23]. The responsibility of establishing these objectives is typically given to unit leaders, who play a critical role in ensuring that each department contributes toward achieving the school's desired outcomes [20].

Third, proposing operational budgets involves the translation of goals and objectives into implementation plans for operational units. These plans are then used to justify financial resource allocation [23,24]. Within a proposed budget, each plan should have detailed cost estimates, allowing school leaders to make informed decisions about how to allocate their funds [24,26].

Fourth, preparing a budget in educational institutions involves determining priorities [22], wherein school leaders are required to rank budgeted items based on their potential impact on the development of the school and their cost-effectiveness [26]. This necessitates ranking each item, based on its ability to achieve the school's goals and objectives [23]. The organization can then allocate more funding toward high-priority items while reducing funding for lower-priority ones.

Finally, establishing a monitoring and evaluating system is crucial for making informed decisions based on data [22]. A possible monitoring and evaluating system could include collecting data on attendance rates, analyzing trends and patterns, and using this information to make informed decisions regarding program adjustments or resource allocation [16]. By regularly monitoring and evaluating the effectiveness of programs or initiatives, schools can ensure that their limited financial resources are being used effectively to achieve their desired outcomes.

### 2.3. Challenges for WSF

Theoretically, by using these five iterative steps of financial resource management, which are commonly employed globally [15,16], schools can support initiatives for sustaining improvement within schools. Nevertheless, in practice, there are many impediments to the effective application of WSF that hamper school innovation [15,17].

#### 2.3.1. Policy Discontinuity

Within the WSF system, potential changes in national policy [27,28] and high staffing costs [6] may inhibit long-term planning and investment in innovation. At the national level, educational reform is often subject to fads and short-term trends [27], forcing school leaders to allocate resources to initiatives that are either temporarily popular or have public support [29]. The lack of a national education policy can also interfere with sustainable school innovation [28]. At the individual school level, the disproportionately high expenditure of WSF funds on staffing costs has hindered the decision-makers' ability to consistently allocate sufficient resources to new initiatives [6]. Therefore, insufficient funds can reduce a school's capacity to continue supporting formerly successful initiatives or policies over time, especially as new initiatives emerge [30].

#### 2.3.2. The Culture of Using All Available Funds

School budgets are often estimated and planned based on projections that may not align with a school's actual expenses or needs [31]. This misalignment can lead to underspending or unused resources at the end of an academic year. Since the WSF claw-back system often motivates schools to spend all available funds before the end of the fiscal year to avoid losing their funding, schools may spend imprudently without proper planning, hindering long-term development [14,32].

#### 2.3.3. Prioritizing Short-Term Objectives

Innovation often requires that several staff members establish a foundation, e.g., acquiring teaching materials or fostering school-community partnerships [33,34]. As funding is limited, schools tend to reward emergent teaching and learning needs that align with departmental and school goals [14,32]. Thus, they may be left with only small budgets for

the subject heads to pilot initiatives [18]. Insufficient funds for launching initiatives also make it challenging for subject heads to execute long-term projects. This can impede the ability of schools to sustain long-term improvements in student outcomes and can also limit schools from undertaking innovative initiatives that contribute to continuous improvement over time.

*2.4. The Principals—The Key Decision-Makers of WSF*

In the school context, it is important to recognize the importance of balancing pedagogical needs with fiscal realities, while ensuring efficient resource utilization, and acknowledge that the establishment of pedagogical standards inherently necessitates consideration of the associated resource requirements and leadership implications [35]. School principals are the key decision-makers regarding funding allocation; i.e., they decide how to allocate funds and oversee their implementation, while ensuring that the funded projects align with school goals [12]. They are also accountable and responsible for their decisions [8,9]. For example, they provide a supportive environment by funding training, facilities, and teaching resources, thereby sustaining daily operations and fostering innovation [10–12].

Studies have noted that various leadership styles can result in different priorities in financial decision-making. For example, research has shown that instructional school leaders can efficiently deploy financial resources to improve a teacher's performance [11]. Similarly, a study on transformational leadership found that transformational school leaders utilize financial resources to support teachers' professional development, enabling them to understand new initiatives that will cultivate positive feelings among teachers about supporting school improvement efforts [10]. Transactional school leaders may use financial resources to provide rewards or incentives to comply with their directives [12].

Recent studies have highlighted that school principals who embody entrepreneurial leadership are particularly adept at utilizing financial resources [35,36]. These leaders leverage their creativity and competencies to generate new value from existing resources, including financial ones [35]. They can bring together disparate resources within their institution and catalyze them in beneficial and innovative ways [36]. The impact of this entrepreneurial approach is profound, leading to the more efficient use of school funds, fostering a conducive learning environment, and driving positive changes within their schools' internal and external environments [35]. This effective resource management enhances the financial sustainability of schools and contributes to their overall educational value and success.

As part of the important role played by school principals, financial decision-making not only pertains to devising optimal budgeting strategies but also to managing the stakeholders involved in the process [8,9]. For instance, to prioritize innovation, principals need to allocate more funding and establish resource allocation committees; however, any reallocation of funds may cause teacher dissatisfaction and undermine the program's success [37,38]. Teachers may raise concerns about prioritizing initiatives if they feel that their own interests are diminished [39]. Thus, schools must sometimes bear the risk of staff morale challenges when sustaining innovative programs [13,37–39].

Drawing insights from the existing literature on principal leadership, it is evident that principals play a critical role in leveraging support for school operations and innovation (see Figure 1). With the allocation of WSF to schools, an iterative process of financial resource management begins. This process is important for deciding how to maintain school operations and sustain innovation. During this process, the principal and teachers-in-charge are key decision-makers. Principals, in particular, wield considerable influence over teachers' perceptions regarding resource allocation. However, studies have tended to overlook how principals make financial resource decisions and tackle the challenges associated with WSF, particularly its impact on the sustainability of school operations and innovation. Therefore, further investigation is warranted to unravel the intricate interplay between principals' perceptions of WSF and the strategies that they employ to promote sustainability in school operations and innovation.

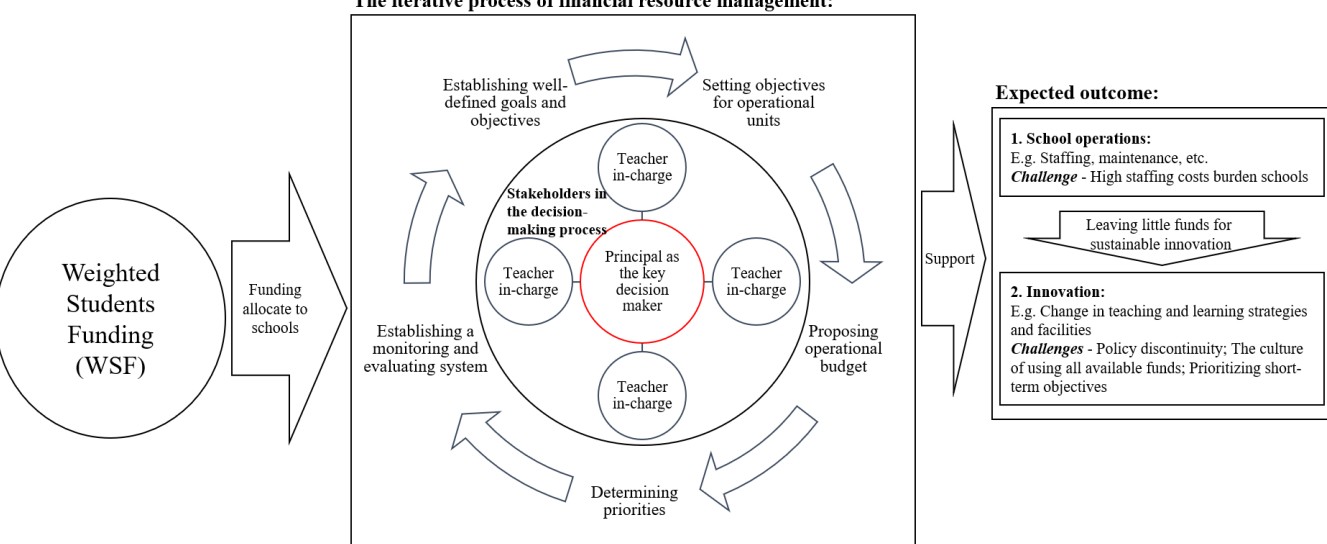

**Figure 1.** A synthesis of the literature reviewed above as a conceptual framework, leading to the research methodology.

### 2.5. Hong Kong Context

The Education Bureau in Hong Kong, the context of this study, has implemented a stringent monitoring system for schools regarding WSF. This approach is consistent with the five steps for managing a school's finances as outlined by [22,23]. In 2002, educational reform was launched in Hong Kong to promote students' whole-person development and lifelong learning through a series of structural, curricular, and pedagogical changes [40]. Although every school in Hong Kong can apply for two million Hong Kong dollars (about USD 260,000) in support from the government to launch innovations that align with government advocacy, initiatives focused on meeting the specific needs of students or teachers are mostly considered to already be funded by WSF. This implies that schools must bear the responsibility of maintaining the sustainability of new projects through the use of their WSF, which tends to increase their financial burden [41]. These challenges highlight the difficulties that Hong Kong principals face in promoting innovation using their WSF. In essence, they must leverage their WSF to achieve their goals, while maintaining educational standards and addressing teachers' concerns regarding resource allocation.

### 3. Methodology

This research adopted a case study approach to address the research questions and the identified gap in the literature. Case studies can provide an in-depth exploration and describe how WSF is used, showing the complex interplay between a principal's strategies and their interactions with teachers, particularly in relation to financial resource management. Such an approach, in situations where the interests of multiple stakeholders need to be balanced, can best capture the dynamic nature of decision-making [42]. Furthermore, the existing literature on financial resource allocation [22,23] and principals' leadership [10,13] highlights a significant gap between theoretical suggestions and authentic application. Despite extensive research in this area, no specific study has provided an in-depth exploration of how principals make sense of leadership and allocate financial resources to achieve the goal of sustainability and innovativeness in schools. We need to explore the complex interplay between the principal's leadership qualities and their ability to navigate the human aspects of financial resource management, especially in relation to teachers. Therefore, using an ethnography study approach will provide a more nuanced understanding of the factors and human interactions contributing to the decision-making process. The findings of this study have the potential to make practical and theoretical contributions by

demonstrating how principals perceive their use of WSF to support daily school operations and encourage sustainable innovation.

*3.1. Sampling*

A principal was selected as the study subject using two selection criteria: (1) the principal had to be utilizing WSF to initiate innovation, with a substantial budget and the support of the school; (2) this innovation had to have been sustained for three years or more. These criteria were used to identify a principal who had effectively addressed the challenge of utilizing WSF to promote sustainable innovation in their school. To facilitate the selection of a suitable candidate, the researcher requested that the Hong Kong Principal Association nominate 10 school principals who fulfilled the selection criteria. The review process involved scrutinizing the schools' plans and records from 2018 to 2022. Three principals were then identified. Two scholars evaluated their school profiles and innovative practices, focusing on school innovation. Then, a site visit to the three schools was conducted by the author. Finally, one principal was selected as the most viable subject of the study. The background of the principal's school and the participant's demographic information are reported in Tables 1 and 2.

**Table 1.** The selected school's background.

| Selected School's Background | Innovation Initiatives and Its Facilities | Sustainability | Principal's Term |
|---|---|---|---|
| Secondary school Background—Over the last fifteen years, the school has faced challenges in attracting sufficient students. Since 2008, the school has struggled to maintain a consistent level of enrollment of students. However, in 2020, the situation changed, and the school began to see an increase in the number of students enrolling in secondary year one. The number of new students enrolling each year has tripled since 2021. As a result of the increase in student intake, the school has also been able to expand its teaching staff from 20 to 36 members. | Experiential learning—Several experiential learning corners (café, "Design Thinking" working space, Gym science, Eco-Lab) | Starting from the academic year 2019/2020 (embedded in design and technology, science subjects, and extra-curricular activities) | Principal on staff since 2018 (5 years) |
| | AI-STEM education—I-Lab STEM room | Starting from the academic year 2020/2021 (embedded in STEM, visual art education, and extra-curricular activities) | |
| | Mindfulness value education—three specific facilities, including a mindfulness room, a mindful gym room, and a petcare room | Starting from the academic year 2021/2022 (embedded in science education and moral and civic education) | |
| | STEM for the community—a new authentic living space for testing students STEM products open to the community | Starting from the academic year 2022/2023 (support AI-STEM education and setting up a community academy for the community) | |

**Table 2.** Overview of the participating staff.

| Participants | Gender | Teaching Experience | Years of Serving in Sample School | Duty at Schools |
|---|---|---|---|---|
| Principal | M | 23 | 5 | Decides school developmental direction and oversees the school's operation |
| Vice-Principal A (Academic) | F | 20 | 6 | Oversees the academic affairs (related to experiential learning and AI-STEM Education) |
| Vice-Principal B (Pastoral care) | M | 18 | 5 | Oversees pastoral care affairs (related to experiential learning and mindfulness values education) |
| Teacher A | M | 18 | 15 | In charge of experiential learning |
| Teacher B | F | 10 | 6 | In charge of AI-STEM education |
| Teacher C | M | 15 | 15 | In charge of mindfulness values education |

*3.2. Data Collection*

The study primarily used semi-structured interviews and school documents (curriculums, lesson plans, and students' work). The author conducted three site visits (240 min) to the school to attend meetings and lessons and engage in conversations with students and teachers. The interviews were conducted with six participants in the school, comprising the school's principal, two vice-principals, and three teachers who were responsible for implementing innovations. Before conducting the interviews, the author reviewed the school's plans and reports in order to comprehensively understand the background of the initiatives, which helped develop the interview questions. The questions probed the participants' general perceptions of using WSF, the school's goals in the past five years, the overall process of assessing financial resource needs, resource allocation decision-making, the various ways of configuring and leveraging resources for innovation, and the strategies used for innovation sustainability. To ensure the credibility and accuracy of responses, the author conducted interviews with the vice-principals and teachers in charge of the initiatives as a form of triangulation [43]. The author also invited the teachers in charge to share their curriculums, lesson plans, and students' work to verify the usage of the new facilities arising from initiatives. During the site visit, the author served as an observer and took field notes to describe the decision-making process in meetings and the learning process during lessons. The author also wrote memoranda after each conservation between the author, teachers, and students. In total, four meetings were attended, and three classes were observed.

*3.3. Data Analysis*

The study adopted the interactive data analysis model described in Ref. [44]. The data analysis process involved two stages. Reflective memoranda were created immediately after conducting interviews and reviewing the relevant documents in the first stage. The author then categorized the data based on budgeting processes, after which data displays were generated to verify the authenticity and validity of the data sources. For example, the author constructed a timeline depicting the budgeting process and then cross-referenced the interview and document data against it, to identify any potentially missing sources of information and verify the validity of the interviewee's claims. This provided a holistic picture of budgeting. To serve as a basis for further data collection, interim summaries for each interviewee were composed. Based on these summaries, data display techniques were used to verify the authenticity of each mentioned event, while examining how the interviewees perceived and interpreted the same situation differently. Any doubtful data were rectified by collecting additional data at the research site.

The author coded the data into descriptive and interpretive categories during the second stage. Descriptive codes were used for definite facts, while interpretive codes uncovered the participants' beliefs behind their actions. Participant feedback enhanced the code's reflexivity [45]. The interviews were conducted in Chinese, with all translated quotations being verified by a professional translator from a reputable newspaper company. Additionally, participants were provided with the opportunity to confirm the accuracy of the translations themselves. Pattern categories were generated using pattern-matching and concept mapping techniques to identify those of the principal's strategies that enhanced innovation sustainability. The author invited an external reviewer, an academic in principal leadership studies, to review and comment on the resulting pattern category analysis, in order to evaluate whether the analysis aligned with the conceptual framework. Finally, three thematic categories emerged in response to the research questions: (1) the principal's perceptions of utilizing WSF; (2) the principal's strategies for leveraging WSF for innovation; and (3) the teachers' reactions.

## 4. Findings

This section presents the main findings regarding the use of WSF over three academic years (2018–2021). The results were categorized according to each academic year, delineat-

ing three distinct themes for each year. Representative quotes are used here to illustrate these themes and highlight the key findings.

*4.1. Gearing up for School Transformation in 2018/19 (Also See Appendix A, Table A1)*

*Theme 1—Perceptions of utilizing WSF: The need to reduce financial rigidity*

The principal recognized the need to move away from the stringent financial management procedures of WSF to encourage equality and innovation in his school. He acknowledged the importance of financial management procedures; however, he said that some procedures were too inflexible and may have hindered creativity. For instance, every extra sum of financial support from WSF required a clear description of the intended use for the money at the beginning of the year, even though the needs of students varied during the year.

> "The government's special funding for new policies, especially for SEN students, is not sustainable in the long term. They may need different experts or equipment to support their learning, and it constantly changes. However, all budgets are fixed at the beginning of the year, most likely spent on hiring teachers." Principal

These rigid processes can restrict a school's ability to adapt to changing student and societal needs, resulting in a lack of innovation. Additionally, the funding formulator of the WSF system can disproportionately affect students and schools from low-income backgrounds, exacerbating the existing inequalities.

> "The current education funding system is unfair and creates inequity. We have many students from low-income families, and the resources we receive aren't enough to properly care for them. We have to put in triple the effort to provide the support they need with the same resources as other schools." Principal

Considering these issues, the principal was committed to promoting equity and innovation by introducing novel and flexible financial management procedures that moved beyond the survival mode cycle. For example, he established a committee to deliberate on the allocation of funds to support students in need.

*Theme 2—Principal strategies: Preparing for student-centered school innovation*

The principal actively worked toward creating alignment among stakeholders to support student-centered innovation as the basis for the following year's changes. First, he evaluated the school's renovation needs through a student-centered lens. He engaged with each student ($n = 150$) individually to understand their expectations, which enabled him to identify those areas needing improvement in order to enhance student retention. Additionally, he guided the middle leaders in drafting departmental plans, ensuring that teachers understood the reasoning behind any changes and the requisite steps involved in implementation. He aimed to train teachers to approach their work more thoughtfully. For example, after the initial draft, the principal met with each department head to discuss their respective plans. Together, they reviewed the plan and clarified every item.

> "It's important for our teachers to clearly understand why they are being asked to do something and the steps involved in implementing it. This helps them to approach their work in a more thoughtful and effective way." Principal

> "Since joining us, he has diligently reviewed our plans, asking questions and seeking to understand our reasoning. He's detail-oriented, even checking our grammar to ensure clarity in our writing." Teacher A

Later, the principal worked with middle leaders, helping them shape the direction of school innovation. Together, they decided on three key directions for the school: for the future, for teachers' and students' well-being, and for the community. With a clear direction in mind, the principal wanted to ensure that the teachers knew the significance of the three key directions by continuously emphasizing the importance of student-centered innovation, even in less relevant scenarios.

> "He talks about his ideal school in every scenario you can imagine, from staff meetings to informal chats to morning assemblies. It's clear that he is passionate about our school." Teacher B

Furthermore, the principal made a commitment to financial resource conservation. He saved money whenever possible and used the remaining funds before the accounting period ended, ensuring the most efficient use of the available resources. For instance, he noticed that the career guidance committee had not spent all of its allocated funds. Upon confirmation, the principal promptly sought quotations for coffee machines, which were to be used for the following academic year's "experiential learning" areas in the café.

> "To prepare for potential fund recalls, I save whenever possible. I'm mindful of my spending, as colleagues often forget their budgets. At year-end, I plan to use the remaining funds effectively before the accounting period ends." Principal

*Theme 3—Teachers' reaction: Recognizing the possibility of imminent change in the school*

Teachers at the school were aware of the possibility of significant changes occurring in the school environment, although they were uncertain about the details. The teachers acknowledged that they learned the importance of meticulous planning and a commitment to consistent improvement, exemplified by their eagerness to revise and enhance their plans multiple times under the principal's guidance. Nevertheless, certain teachers found it hard to understand the principal's operational plan, even if they agreed with the school's development direction, leaving them feeling unsure and waiting for more explicit instructions.

> "At first, all I knew was his slogan ('Three key directions for the school'). I agreed with his directions but I didn't clearly understand what he wanted to do. It felt like waiting for the thunder before the rain." Teacher A

Notably, many teachers were initially oblivious to the principal's efforts to conserve funds and only became aware of it when the plans for constructing a cafe were announced for the upcoming academic year.

> "It finally dawned on us that he had saved a lot when he announced plans to build a cafe in the upcoming academic year." Vice-Principal A

*4.2. Seeding Innovation through Resource Optimization in 2019/20 (Also See Table A2)*

*Theme 1—The perception of utilizing WSF: Adversity breeds survival intention*

The principal perceived the challenges of the limited financial resources facing the school, which prompted him to reflect on how to ensure the school's survival and future success. To initiate innovations within the school, he relied on WSF for seed funding. This limited funding required him to consider the most effective use of resources.

> "I only have a small saving from the WSF, so I need to use it effectively to make some changes. At least . . . If you can't succeed, you should aim for a noble failure." Principal

The principal also understood that he had to seek creative ways to meet the student's needs and improve the school's overall performance by exploring non-monetary resources.

*Theme 2—Principal strategies: Practice as a champion*

The principal took a hands-on approach to collaborating with colleagues, from the planning stage to the actual construction of facilities. Recognizing the limited budget for initiatives, he prioritized spending on "hardware" (facilities) since he foresaw the students' need for authentic experiential learning and gathering places.

> "Due to limited resources, I decided to prioritize investing in facilities over the curriculum. I believe that improving our facilities can create a more positive learning environment for our students." Principal

He then encouraged the free flow of ideas among his team to achieve the best outcomes. Together, they devised a plan to use their limited money to support experiential learning development by building a café, designing a thinking workspace, "gym science" area, and an eco-lab. To address any shortfall in financial resources, the principal purchased materials from an online store and contacted skillful janitors and colleagues, convincing them to remain after school to perform the renovations. Thus, they completed the project at a much lower cost than if they had hired a contractor.

> "It's unbelievable that the principal and janitor used a big hammer to break down the wall and built the cafe by themselves, purchasing all the necessary materials." Vice-Principal B

*Theme 3—Teachers' reactions: Unleashing the innovative potential of teachers*

The teachers said that the principal removed those barriers that were impeding their ability to innovate. The availability of new facilities served as a catalyst for stimulating pedagogical change and sparking innovative thinking among the teachers. This encouraged the teachers to design a physical space that promoted experiential learning and pushed the students to reach their full potential. Some teachers began thinking about developing a curriculum that would effectively use these new facilities and sought to design innovative projects to improve their students' quality of education.

> "The new facilities have inspired me to think outside the box and design a learning environment that encourages experiential learning and pushes students to reach their full potential." Teacher A

Notably, despite allocating most of the funding toward these new initiatives, the teachers reported that the principal did not reduce the daily operational expenses of any department, ensuring that stable financial support was provided to all areas of the school.

> "Despite implementing numerous new initiatives, all budget plans were approved under his leadership." Vice-Principal A

*4.3. Scaling up Innovative Practice in 2020/21 (Also See Table A3)*

*Theme 1—The perception of utilizing WSF: Confidence in moving away from reliance on WSF*

The principal was confident in the school's ability to identify and secure alternative sources of funding to support their future initiatives, moving away from relying solely on WSF. His plan involved using WSF to sustain four new facilities for experiential learning, as the expenses were reasonable and well worth the investment in providing innovative educational opportunities for their students.

> "We have four new facilities for experiential learning that we sustain through using WSF. The expenses are reasonable and well worth the investment in providing innovative educational opportunities for our students." Principal

The principal claimed that the new facilities demonstrate the potential of the school's financial management and its innovative teaching methods, which will help them secure external financial support for future initiatives.

> "I have four successful cases that we can use as examples to secure external financial support for future initiatives, showing that we can pursue innovative funding opportunities and reduce our reliance on WSF." Principal

*Theme 2—Principal strategies: Empowering teachers to become proactive innovators*

The principal played an important role in empowering teachers to become proactive innovators by supporting them in seeking the resources necessary for implementing their own initiatives. The principal began by showcasing the school's facilities and experiential learning to external parties, establishing connections, and broadening funding sources beyond their reliance on WSF.

"Our principal's efforts to promote our school's success stories, particularly our achievements in student learning and innovative facilities, have attracted the interest of external parties who have expressed a desire to offer additional financial support." Teacher B

Meanwhile, the principal collaboratively established the expected student learning outcomes with colleagues who were interested in initiating the new projects. The principal then encouraged those teachers to apply for funding opportunities that would support future innovative teaching practices and the expected learning outcomes. He guided and reviewed the teachers' proposals, to ensure that they aligned with the school's academic direction.

"He provides direction and guidance, allowing us to develop proposals based on our own ideas. He reviews and evaluates our proposals to ensure alignment with his vision for the school's growth and sustainability." Teacher C

*Theme 3—Teachers' reaction: Teacher-led resource mobilization*

Since the beginning of the 2021–2022 academic year, several teachers had sought out resources to improve student growth and development. These teachers placed their trust in the principal, due to his successful track record of transforming limited resources into radical changes in the school. His success and the strong networks he established with potential funders resulted in widespread enthusiasm among other teachers regarding applying for funding for AI-STEM education.

"After successfully securing AI-STEM funding, three more colleagues approached us to help write a funding proposal for mindfulness education, which we were able to secure the following year for a total of seven million dollars." Vice-Principal B

The teachers' efforts were reinforced by the allocation of HKD 3 million in grants (USD 450,000) to the school to develop its AI-STEM program and STEM Lab. The grants were received due to external funding gained from charity foundations. This funding resulted in new prospects for groundbreaking projects and ongoing progress and development in the school, resulting in three additional funding opportunities for mindfulness education, totaling HKD 7 million, planned for 2021/22.

The school's new entrepreneurial spirit did not appear to incite any resentment in the teaching staff. Teachers reported that anyone could apply for funding and receive their earned resources. Those who did not aim to initiate any projects did not appear to object to the distribution of resources.

"There is no resentment or jealousy over resource allocation. Everyone has the opportunity to apply for funding and earn what they work for. Don't forget that it also implies workload." Teacher A

## 5. Discussion

A school's sustainability is inherently interwoven with innovation; this is the engine that drives progress and adaptability in an ever-evolving educational landscape [7]. Innovation in teaching and learning is a deliberate craft honed by school principals who strategically allocate their budgets. By investing in innovative practices, principals sow the seeds for a vibrant, future-ready learning environment, thereby ensuring the long-term sustainability of their schools.

The study's findings reveal that the principal played a crucial role in driving innovation using WSF. The principal's perception of WSF utilization was largely influenced by his experiential encounters and challenges in managing a meager budget while striving to promote equity and innovation in his school. The primary concern about WSF primarily revolves around its inflexibility and inadequacy. The rigid administrative procedures associated with WSF appear to significantly impede creativity in catering to students' needs. WSF policies regarding allocating resources to schools often exacerbate the existing disparities, favoring wealthier schools. Despite the principal using WSF as the primary

source for initiating innovations in his school in the second year, the credit also goes to the principal's strategies for saving resources. The principal stated that WSF did not adequately provide for resources to address diverse student needs. He claimed that WSF is best suited for a limited set of needs, such as facilitating general school operations, piloting initial innovations, and supporting the operational expenses of innovation. Thus, to enable schools to promote sustainable innovation, there is an urgent need to move beyond a reliance on WSF.

The principal demonstrated layered leading strategies in managing and allocating his budget to maintain school operations and promote innovations. In the first year, the principal focused on formulating a consensus among teachers and students regarding the school's development direction. This was achieved by enhancing the teachers' planning capacity, consulting with students, and sharing educational goals with teachers. Building on this foundation, the principal encouraged innovation and exhibited infectious enthusiasm for bringing the initiatives to fruition in the second year. By strategically planning initiatives with core team members and being involved in every working process, he was able to actualize the experiential learning initiatives with a limited budget while passing on his passion to the teachers. The renovations served as a symbolic catalyst for subsequent changes. In the third year, the principal shifted his role from being a facilitator to mainly focusing on seeking funding for teachers with innovative ideas. He continuously acknowledged the teachers' contributions and guided colleagues to further elaborate upon their ideas, ultimately assisting them in compiling proposals for funding applications. Throughout the three years, the principal's strategies involved scaling up his influence by practicing what he preached. Notably, the principal never raised the issue of financial difficulties with the teachers but instead utilized school savings to actualize his innovative ideas while seeking potential funders to support the teaching staff.

This case study underscores the important role of principals' entrepreneurial behaviors in promoting the effective utilization of resources and driving positive change in schools [19,46]. The principal in this case study has effectively embodied entrepreneurial leadership, aligning with the desired characteristics highlighted in the literature [35,36]. The principal demonstrates the ability to leverage creativity and competencies to generate new value from existing resources, including financial ones, as seen in the strategic allocation of a limited budget to foster experiential learning initiatives and school renovations. Such a proactive approach to seeking resources, building consensus among staff, recognizing innovative opportunities, and defending new initiatives mirrors entrepreneurial traits [19]. These behaviors have led to the efficient use of school funds, a conducive learning environment, and significant positive changes within the school's internal and external environments.

Teachers underwent role changes, moving from being learners to collaborators and, thence, to leaders over the three years. In the first year, the teachers acted as learners who agreed with the school's development direction, while awaiting radical changes. They demonstrated their eagerness to revise and enhance their plans multiple times under the guidance of the principal. In the second year, the availability of new facilities served as a catalyst for pedagogical change and sparked innovative thinking among the teachers, encouraging them to become collaborators in designing an environment that promoted experiential learning. In the third year, the experience of experiential learning resulted in widespread enthusiasm among the teachers toward becoming leaders, as they composed proposals to apply for funding for various initiatives. Despite limited financial resources in the school, there were no criticisms regarding funding allocation. Most teachers were unaware of the straitened fiscal position of the school and the criteria used by the principal to allocate funds to new innovations. Despite the availability of the relevant information, many teachers seemed oblivious to these financial matters.

This study delves into the principal's perception of utilizing WSF, the strategies employed, and the teachers' reactions, thereby unveiling the challenge of school sustainability in both operational and innovative aspects. The findings suggest that WSF, in its current

form, tends to obstruct rather than aid school sustainability. However, this study also casts a spotlight on the pivotal role of principal leadership, underscoring the necessity for principals to embody entrepreneurial leaders, inspiring and spearheading innovation in their schools amid constraints such as rigid funding structures similar to that of WSF. The principal in the study exemplifies this by effectively leveraging his resources, both human and financial, to cultivate an environment that is conducive to innovation and learning. This successful exploitation of resources represents teachers' professional capital growth [47]. The principal built this capital through various means: fostering consensus among staff (social capital), identifying innovative opportunities (decisional capital), championing new initiatives (decisional capital), and proactively sourcing resources (human capital). Under his leadership, teachers transformed from learners to collaborators and, ultimately, to leaders. This metamorphosis was powered by a shift in pedagogical approach and the introduction of new facilities that encouraged experiential learning. Therefore, despite the challenges posed by limited and inflexible funding mechanisms, the proactive and strategic use of resources can bolster teachers' professional capital and school sustainability by fostering innovation.

*5.1. Theoretical Implications*

The study's findings regarding the principal's perceptions of WSF align with the existing literature from beyond Hong Kong, highlighting this school's tendency to allocate a significant portion of its budget to staffing and maintenance [6]. However, the participating school's staff did not appear to have any concerns about prioritizing initiatives. In reality, the principal used WSF as the primary source for piloting and supporting the daily operational costs associated with initiatives. While implementing such initiatives required significant effort to save resources, the findings suggest that WSF was adequate for sustaining the school's innovative endeavors, even if it was not completely sufficient for radical changes.

This study adds value to the existing literature on principal leadership by shedding light on how principals can utilize WSF and government-regulated financial management mechanisms to foster teacher growth. Prior research has largely focused on leaders' innovations and how they manage their budgets [8,9]. The findings of the present study show how a principal tactfully used regulated financial management mechanisms to encourage teachers to initiate projects according to his example initiatives. This finding indicates that the principal was able to expand the management of WSF beyond the typical budgeting needed for the school's day-to-day operations, and this led to some teachers acquiring the energy and skills to advance their own initiatives.

The findings also demonstrate how principals encourage consensus-building and advocacy for innovation, thereby increasing innovative practices through the effective use of financial resources. This study's findings unlock new avenues for financial resource management studies as principals not only comply with regulations but also devise ways to conserve and acquire funding for school initiatives. Future studies should investigate how principals use their budgets creatively to foster and sustain innovation.

The principal in this study has demonstrated the potential of entrepreneurial leadership in successfully addressing financial challenges. However, it is important to recognize that the ability to manage financial resources effectively is unique to each individual. This distinctive experience illuminates an alternative approach to leadership in financial management, encompassing responses to planned and emergent situations, among other facets that are specific to administration. Studies suggest that school leaders who adopt instructional, transformational, or transactional styles can also effectively allocate financial resources to enhance teacher performance [10–12]. Given the broad range of circumstances and leadership styles, directly comparing their experiences would be inappropriate.

### 5.2. Practical Implications

WSF initially played a significant role in promoting small-scale innovation. However, afterward, its focus shifted toward sustaining innovation so that it became a regular part of school operations. Such sustainment was revealed by the incorporation of new facilities into the experiential learning curriculum by the teachers, with support from WSF. Studies have mainly pointed to the limitations of WSF in spurring radical initiatives [18]. However, the present findings indicate that while WSF may have a limited ability to support radical initiatives, the effective allocation of financial resources can still enable sustained innovation. School leaders can guide teachers in embedding innovations into daily operations to ensure the possibility of using WSF to maintain innovation sustainability.

Unlike previous findings showing that teachers are often dissatisfied with school funding decisions (REFs), the present study findings revealed that many teachers were relatively indifferent toward financial matters. This attitude appears to have facilitated the smooth allocation of financial resources for new initiatives. Future studies should explore the cause and other potential influences of this issue.

### 5.3. Policy Implications

The principal's perceptions of WSF reveal a concerning trend toward exacerbating school inequity, which seriously threatens the sustainability of school improvement. Schools operating under such inequitable conditions often struggle to maintain their educational standards and innovate, thereby undermining their long-term viability. This finding underscores an important issue regarding WSF policy. Schools that serve students with special needs require greater resources to provide an equitable education. Despite this need for additional resources, typical funding allocation practice is solely based on the number of students and, thus, fails to address the unique needs of each school community; therefore, schools with greater needs due to their student profiles often receive insufficient funding, further perpetuating pre-existing educational inequities while negatively impacting such a school's performance and its ability to attract students and parents. This cycle of inequity and underperformance erodes schools' sustainability, impairing their capacity to evolve and adapt to changing educational landscapes. Accordingly, there is a pressing need for a more equitable approach to funding education that prioritizes the diverse needs of students and schools. This could involve adopting a needs-based funding formula and providing targeted support for smaller schools that promotes equitable resource allocation and successful outcomes for all students, ultimately reinforcing school sustainability in terms of continuous improvement.

The study's findings underscore the important role of financial flexibility, particularly in terms of savings, in fostering the initiation of new initiatives. This heightened flexibility directly contributes to the sustainability of school improvements. However, the "call-back" system, as described by the author of [14], limits the potential generation of the additional funds required to pilot these initiatives. Extending the duration of the "call-back" system from a single financial year to two or more years may facilitate schools in retaining more resources, enabling them to pilot new initiatives or enhance existing practices. This potential change could significantly contribute to the sustainability of initiative operations and continuous school improvement processes. To mitigate the risk of misappropriating funds to support untested innovations, the effective management of financial resources in schools is of paramount importance. To this end, the authors of Refs. [22,23] propose a five-step iterative process that educational authorities and school principals should implement.

## 6. Limitations and Conclusions

As is typical of case study research, the findings of this study cannot be generalized. One fundamental constraint in studying school finances is the sensitive nature of the associated data. Financial information, especially within public schools, is often regarded as confidential, thereby limiting public accessibility. Schools seldom disclose their financial

statements, as such disclosure may infringe upon their privacy rights. This lack of transparency renders the acquisition of comprehensive and reliable data challenging, creating a significant hurdle for research in this area. However, this research aims to address a question that has seldom been explored—the use of WSF to spur innovation in schools. Financial management in schools is often opaque and is seldom publicly disclosed or discussed in the literature. The school was purposely selected because it had previously struggled with WSF, yet it overcame these challenges while using WSF to foster innovation for its students. By conducting a case study, the author was able to investigate the interactions between the school's principal and its teachers in depth, focusing on gaining a deep understanding of the dynamic changes in the school's culture and budgeting practices. This approach is particularly valuable, as many schools are currently facing challenges with their own WSF. Another limitation concerned the length of the study. Although three years may appear sufficient for understanding the impact of a funding decision, ideally, a longer period would have provided better evidence of the sustainability of the initiatives. Our study offers insights that can advance the knowledge of principals and other stakeholders, exploring the possibility of using various leadership strategies to leverage WSF for school innovation and enhance school sustainability.

In conclusion, this case study has revealed the significant impact that a principal's character, in this case, his entrepreneurial traits, can have on a school's ability to effectively use its WSF for maintaining operations and fostering innovation. By actively engaging with staff, advocating for innovation, seeking resources, and mitigating risks, the principal was able to transform a failing school into an innovative educational establishment [35]. These findings align with previous studies on teachers' entrepreneurial behavior [19,46], which suggest that individuals with entrepreneurial characteristics tend to enact a series of competencies and possess attributes that enable them to seize opportunities to scale up innovation in schools. This not only elevates the quality of education but also significantly contributes to the sustainability of schools by enabling them to adapt and thrive in an ever-changing educational landscape. Consequently, the author argues that a deeper understanding of the entrepreneurial behaviors exhibited by school leaders is crucial for promoting the effective utilization of WSF and driving positive change to ensure school sustainability through continuous improvement.

**Funding:** This research received no external funding.

**Institutional Review Board Statement:** The animal study protocol was approved by the Institutional Review Board (or Ethics Committee) of The Education University of Hong Kong (protocol code 2022-2023-0335 and 16/05/2023) for studies involving animals.

**Informed Consent Statement:** Informed consent was obtained from all subjects involved in the study.

**Data Availability Statement:** Not applicable.

**Conflicts of Interest:** The authors declare no conflict of interest.

## Appendix A

**Table A1.** Findings of the Year 2018/19.

| Findings | Pattern Category | Representative Quotes |
|---|---|---|
| The need to reduce financial rigidity | Agreeing with the need for stringent financial management practices | "As schools, we're accountable to society for how we use taxpayer resources. To make responsible decisions, we follow financial management requirements (five steps) by Education Bureau and align our school plan with our goals to ensure transparency and accountability" Principal |
| | Exacerbating inequity for struggling students and schools | "The current education funding system is unfair and creates inequity. We have many students from low-income families, and the resources we receive aren't enough to properly care for them. We have to put in triple the effort to provide the support they need with the same resources as other schools." Principal |
| | Rigidity leads to innovation discontinuity | "The government's special funding for new policies, especially for SEN students, is not sustainable in the long term. They may need different experts or equipment to support their learning, and it constantly changes. However, all budgets were fixed at the beginning of the year, most likely spent on hiring teachers." Principal |
| | Breaking free from the vicious cycle of survival | "The current school funding system creates a vicious cycle of survival where weaker schools like ours are losing resources while richer schools get more. We can't keep going like this if we want to provide the best education for our kids. The system needs to change, and that's something I have felt strongly about since my first year here." Principal |
| Preparing for Student-Centered School Innovation | Evaluating school renovation needs through a student-centered lens | "In my first few months, I saw that the school was in disarray with poor maintenance, which affected student retention. So, I spoke with each student individually to understand their expectations of me." Principal |
| | Comprehensive feedback on departmental plans to establish a clear "why" and "how" | "It's important for our teachers to clearly understand why they are being asked to do something and the steps involved in implementing it. This helps them to approach their work in a more thoughtful and effective way." Principal "Since joining us, he has diligently reviewed our plans, asking questions and seeking to understand our reasoning. He's detail-oriented, even checking our grammar to ensure clarity in our writing." Teacher A |
| | Engaging middle leaders to shape school innovation direction | "He reached out to me, [Vice-Principal B], and some of our passionate colleagues to discuss the school's future direction. Over three months, we had many conversations where he shared his insights on what he had learned from students and his thoughts on the role of schools. Together, we decided on three key directions: our schools are for the future, for the well-being, for the community." Vice-Principal A |
| | Reinforcing the importance of a student-centered innovation direction to teachers | "He talks about his ideal school in every scenario you can imagine, from staff meetings to informal chats to morning assemblies. It's clear that he is passionate about our school." Teacher B |
| | Committing to financial resource conservation | "To prepare for potential fund recalls, I save whenever possible. I'm mindful of my spending, as colleagues often forget their budgets. At year-end, I plan to use the remaining funds effectively before the accounting period ends." Principal |
| Recognizing the possibility of imminent change in the school | Recognizing the significance of rigorous planning and committing to continuous improvement | "When he asked me to start with the "why" of each item, it was a new way of thinking. This led me to polish my plan more than three times under his instruction, seeking to refine and improve it each time." Teacher C |
| | Anticipating change with a clear sense of purpose | "At first, all I knew was his slogan. I agree with his direction but I didn't clearly understand what he wanted to do. It felt like waiting for the thunder before the rain." Teacher A |
| | Unawareness of the principal's fiscal conservation | "It finally dawned on us that he had saved a lot when he announced plans to build a cafe in the upcoming academic year." Vice-Principal A |

**Table A2.** Findings of the Year 2019/20.

| Findings | Pattern Category | Representative Quotes |
| --- | --- | --- |
| Adversity breeding survival intention | Reliance on WSF as seed funding for innovation initiation | "I only have a small saving from WSF, so I need to use it best to do some changes. At least ... If you can't succeed, you should aim for a noble failure." Principal |
| | Actively seeking non-monetary resources | "I have no expectations on how WSF can help us, so I focus on exploring non-monetary resources within the school to achieve our goals." Principal |
| Practice as a champion | Prioritizing infrastructure over the curriculum | "Due to limited resources, I have decided to prioritize investing in facilities over the curriculum. I believe that improving our facilities can create a more positive learning environment for our students." Principal |
| | Inclusive and supportive decision-making | "I invite Vice-principals and several teachers to determine the priority of building facilities that would be highly beneficial. We meet every day before it starts." Principal<br>"He prioritizes the best interests of the students and reminds us not to be overly concerned with financial constraints. He just encourages the free flow of ideas to achieve the best outcomes." Teacher A |
| | Networking with potential colleagues | "It's unclear how the principal found out about our electrical worker licenses, but he proactively reached out to us to collaborate on plans for building a new cafe on campus." Teacher A |
| | Exemplifying self-sacrifice | "It's unbelievable that the principal and janitor used a big hammer to break down the wall and built the cafe by themselves, purchasing all the necessary materials." Vice-Principal B |
| Unleashing the innovative potential of teachers | Facilities acting as a catalyst | "After seeing the new facilities, we better understood the school's development direction and how we can better serve our students." Teacher C |
| | Stimulating pedagogical change | "The new facilities have stimulated us to reevaluate our teaching methods and explore innovative ways of providing authentic experiential learning opportunities for our students." Teacher B |
| | Sparking innovative thinking for the next level | "The new facilities have inspired me to think outside the box and design a learning environment that encourages experiential learning and pushes students to reach their full potential." Teacher A |
| | Avoiding influencing departmental financial expenses | "Despite implementing numerous new initiatives, all budget plans were approved under his leadership." Vice-Principal A |

**Table A3.** Findings of the Year 2020/21.

| Findings | Pattern Category | Representative Quotes |
| --- | --- | --- |
| Confidence in moving away from a reliance on WSF | Supporting the sustainability of past innovations | "We have four new facilities for experiential learning, which we sustain through using WSF. The expenses are reasonable and well worth the investment in providing innovative educational opportunities for our students." Principal |
| | Striving for financial self-sufficiency to fuel innovation | "I have four successful cases that serve as examples of securing external financial support for future initiatives, showing that we can pursue innovative funding opportunities and reduce our reliance on WSF." Principal |

**Table A3.** *Cont.*

| Findings | Pattern Category | Representative Quotes |
|---|---|---|
| Empowering teachers to become proactive innovators | Networking with potential funders | "Our principal's efforts to promote our school's success stories, particularly our achievements in student learning and innovative facilities, have attracted the interest of external parties who have expressed a desire to offer additional financial support." Teacher B |
| | Collaboratively establishing expected student learning outcomes | "Our principal encourages us to build upon our success in utilizing our innovative facilities and think about the next steps for our school's growth. He acknowledges and encourages us to pursue new innovations and define the expected outcomes for our students in the future." Vice-Principal B |
| | Encouraging teachers to apply for funding | "He provides direction and guidance, allowing us to develop proposals based on our own ideas. He reviews and evaluates our proposals to ensure alignment with his vision for the school's growth and sustainability." Teacher C |
| Teacher-led resource mobilization | Teachers placing their trust in the principal | "He is right. We successfully secured external funding sources for our mindfulness education, which support renovating a new special room and employing additional staff for curriculum development." Vice-Principal A |
| | Inspiring a plan-writing culture among teachers | "After successfully securing AI-STEM funding, three more colleagues approached us to help write a funding proposal for mindfulness education, which we were able to secure the following year for a total of 7 million dollars." Vice-Principal B |
| | The absence of complaints regarding merit-based funding allocation | "There is no resentment or jealousy over resource allocation. Everyone has the opportunity to apply for funding and earn what they work for. Don't forget that it also implies workload." Teacher A |

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
