# Peer review of "Unpacking the Principal Strategies in Leveraging Weighted Student Funding"

_sustainability, doi:10.3390/su151612592_

Round 1

Reviewer 1 Report

Weighted Student Funding (WSF) is an important topic. The selected case (case study approach) is an very interesting research.

In the overall view, he article is well-structured. The text is clear and precise.

In the introduction, little more information about the study should be added (1 - 2.5).

If useful (I think it would be useful), a subchapter with more insights into current research (regarding WSF) could be added (e. g. Tuchmann et al., 2022).

It would be desirable to explain the figure (p. 5).

The method is clearly described (maybe more material could be added in the appendix, e.g. excerpt raw data/code system).

Findings and discussion, limitation and conclusion are well formulated.

Author Response

Reviewer 1

  1. Weighted Student Funding (WSF) is an important topic. The selected case (case study approach) is an very interesting research.

Response: Thanks for noting our careful choice of sample.

  1. In the overall view, the article is well-structured. The text is clear and precise.

Response: Thank you so much.

  1. In the introduction, little more information about the study should be added (1 - 2.5).

If useful (I think it would be useful), a subchapter with more insights into current research (regarding WSF) could be added (e. g. Tuchmann et al., 2022).

Response: Thank you for your suggestions. I have revised the introduction section to provide a clearer context for our study. Additionally, I've also updated the literature review within the section on principals. Specifically, I've focused on expanding our discussion on entrepreneurial principals, drawing on recent research and emerging trends in the field.

  1. It would be desirable to explain the figure (p. 5).

Response: Thank you for your suggestion. I try to further describe it.

  1. The method is clearly described (maybe more material could be added in the appendix, e.g. excerpt raw data/code system).

Response: Thank you for your insightful recommendation. I have reinstated the code, which is now included in the appendix. I am confident that this addition will offer a more comprehensive understanding of its relevance and detailed application within the context of our current research.

  1. Findings and discussion, limitation and conclusion are well formulated.

Response: Thank you for your acknowledgment. It's an honor to gain insights from your comments.

Reviewer 2 Report

The topic of the study is not new, but given the content and methodological revolution taking place in education, it is timely. The financing of the transition can be based on increasingly limited resources. Therefore, the WSF plays a significant role in the financing of autonomous school decisions. However, an important condition for this role is that WSF spending is based on competent managerial decisions. The authors examine these decisions. Their method is to present the strategy of a successful director based on three well-formulated research questions. They investigate the director's opinion on the task of the WSF, the strategy for the use of funding, as well as the opinions of colleagues in this regard. The topic is presented in an appropriate and relevant literature review, mostly referring to recent sources. It would have been worth referring to:

Demirbilek & Çetin (DOI: 10.33200/ijcer.847110), or Brauckmann-Sajkiewicz & Pashiardis (DOI: 10.1080/13603124.20202418046).

The selection of the interviewee was done in a thorough and careful manner, also assessing the environment. The theoretical description of the interactive data analysis model (row 266) and the justification for its application in the present research should be more detailed for the reader who is not familiar with the procedure.

The structuring of the results according to school years and the clear presentation of the three topics within the years are easy to understand and clear. With this method, the development of research topics over time can be followed. The quoted interview excerpts well support what the authors wish to state. Summary of results should not go into the discussion. In addition, not enough references (three in total) are made to the results of other authors (rows 570 and 581). It should present the positions of several researchers and compare them with the authors’ ones. The limitation of the article also ensures that the author highlights the importance of the entrepreneurial approach as an essential finding from the results, but due to its lack of methodological foundation, I do not recommend this manuscript to publication.

The authors use the American English in a consistent manner. The authors' language use contains only minor typos and mainly uses the academic style throughout the entire paper. Proof-reading by a native English is advisable.

Author Response

I wish to thank the anonymous reviewer for his/her careful reading of the manuscript and helpful comments. I believe I have used these comments in a straightforward way to improve the research. My detailed responses to the review comments are presented in a point-by-point format. Related changes are highlighted in red in the main text.

Reviewer 2

  1. The topic of the study is not new, but given the content and methodological revolution taking place in education, it is timely. The financing of the transition can be based on increasingly limited resources. Therefore, the WSF plays a significant role in the financing of autonomous school decisions. However, an important condition for this role is that WSF spending is based on competent managerial decisions. The authors examine these decisions. Their method is to present the strategy of a successful director based on three well-formulated research questions. They investigate the director's opinion on the task of the WSF, the strategy for the use of funding, as well as the opinions of colleagues in this regard. The topic is presented in an appropriate and relevant literature review, mostly referring to recent sources. It would have been worth referring to:

Demirbilek & Çetin (DOI: 10.33200/ijcer.847110), or Brauckmann-Sajkiewicz & Pashiardis (DOI: 10.1080/13603124.20202418046).

Response: Thank you for your insightful suggestion. I have carefully reviewed the readings you recommended, and I found them to be extremely beneficial. I have incorporated these sources into the literature review and made further modifications to the discussion sections. Your input has significantly enriched the depth and breadth of my research.

  1. The selection of the interviewee was done in a thorough and careful manner, also assessing the environment. The theoretical description of the interactive data analysis model (row 266) and the justification for its application in the present research should be more detailed for the reader who is not familiar with the procedure.

Response: Thank you for your feedback. I'm addressing your comment in conjunction with Comment 3 for efficiency. I believe that including an appendix will provide greater clarity for the readers. I appreciate your guidance in this matter.

  1. The structuring of the results according to school years and the clear presentation of the three topics within the years are easy to understand and clear. With this method, the development of research topics over time can be followed. The quoted interview excerpts well support what the authors wish to state. Summary of results should not go into the discussion. In addition, not enough references (three in total) are made to the results of other authors (rows 570 and 581). It should present the positions of several researchers and compare them with the authors’ ones. The limitation of the article also ensures that the author highlights the importance of the entrepreneurial approach as an essential finding from the results, but due to its lack of methodological foundation, I do not recommend this manuscript to publication.

Response: Thank you for your insightful recommendation. I have reinstated the code, which is now included in the appendix. I am confident that this addition will offer a more comprehensive understanding of its relevance and detailed application within the context of our current research.

On the other hand, I appreciate your feedback regarding the insufficient emphasis on the importance of the entrepreneurial approach and your recommended reading materials. However, this special issue is focused on the use of financial resources for improving school sustainability. Thus, I've chosen to remain within this scope and discuss the authentic practices of the principal in managing financial resources.

Your insights are valuable and highlight the potential to construct a theoretical framework centered on entrepreneurial leadership's role in using financial resources in the future. As such, I've added a paragraph about entrepreneurial leadership and explored the relationship between the study findings and entrepreneurial leadership in the discussion section (see new paragraphs). This addition aligns with my response to your first comment, and I believe it enhances the depth and relevance of the manuscript.

  1. The authors use the American English in a consistent manner. The authors' language use contains only minor typos and mainly uses the academic style throughout the entire paper. Proof-reading by a native English is advisable.

Response: Thank you for your suggestions. I invited a retired professor from the English department to review my writing. The credit should go to him.

Reviewer 3 Report

Comments

The manuscript “Unpacking the Principal Strategies in Leveraging Weighted”

An important limitation and something mentioned by the researcher of the Case Study methodology is the possible bias because the researcher is the one who specifies the phenomenon to be studied, chooses the theoretical aspects and background, weighs the relevance of the different sources, and analyzes the causal relationship between events. Apparently, the writing leads to highlight the leadership and administrative skills of a school when addressing a principal and teachers. However, using a good methodology that is replicable would give greater objectivity.

In this case and as indicated by the authors, its replication is not possible, due to the condition of the methodology and the experience that is very particular to the school and the participants that were approached regarding Weighted Student Funding (WSF).

The experience and skills in managing funds is a characteristic that depends on individual abilities. A study that contrasts this experience with others could provide what makes a way of leading differently, managing funds, responding to emergent situations not initially considered in a plan, and others that are specific to the administration.

On the other hand, the circumstances are so dissimilar that it could not be said if this or another experience is better.

When reading the research questions, the conclusions do not account for your answer. On the one hand, experience is addressed, but the question is perception. Although this information and the strategies are collected throughout the manuscript, at the end in the conclusion it is not finally understood what the answer is and that is due to the variability of response and even improvisation in the face of some problems, so to speak, which makes A scenario at this school would not necessarily be at another school.

I am not sure if financial flexibility is adequate for discretionary application purposes as this could lead to misuse of funds on innovations that have not been tested to ensure successful application. For this reason, the initial plans, and the follow-up of these are relevant to be able to have clarity on the use of the funds. In addition, responding to emergent situations in a context of limited funds could affect those aspects that impact but are longer term.

Author Response

I wish to thank the anonymous reviewer for his/her careful reading of the manuscript and helpful comments. I believe I have used these comments in a straightforward way to improve the research. My detailed responses to the review comments are presented in a point-by-point format. Related changes are highlighted in red in the main text.

Reviewer 3

  1. An important limitation and something mentioned by the researcher of the Case Study methodology is the possible bias because the researcher is the one who specifies the phenomenon to be studied, chooses the theoretical aspects and background, weighs the relevance of the different sources, and analyzes the causal relationship between events. Apparently, the writing leads to highlight the leadership and administrative skills of a school when addressing a principal and teachers. However, using a good methodology that is replicable would give greater objectivity.

In this case and as indicated by the authors, its replication is not possible, due to the condition of the methodology and the experience that is very particular to the school and the participants that were approached regarding Weighted Student Funding (WSF).

Response: Thank you so much for pointing out this issue. In order to provide the reader with a full understanding of the limitations of my study on school finance, I have further enriched the limitations paragraph. I try to emphasize the challenge of comprehensively reviewing school financial information across contexts, which connects to why I chose a case study methodology. Focusing in-depth on one district provides more nuanced insights that may be transferable to other similar contexts. These additions help convey a balanced perspective on the constraints of this research while justifying the value of the approach.

  1. The experience and skills in managing funds is a characteristic that depends on individual abilities. A study that contrasts this experience with others could provide what makes a way of leading differently, managing funds, responding to emergent situations not initially considered in a plan, and others that are specific to the administration.

On the other hand, the circumstances are so dissimilar that it could not be said if this or another experience is better.

Response: Thank you for your suggestions. I have taken reviewer 2's advice to narrow the focus to entrepreneurial leadership. Therefore, I have added one paragraph in the literature review and discussion sections to achieve this. I have also tried integrating your insightful suggestions into the theoretical implications section.

  1. When reading the research questions, the conclusions do not account for your answer. On the one hand, experience is addressed, but the question is perception. Although this information and the strategies are collected throughout the manuscript, at the end in the conclusion it is not finally understood what the answer is and that is due to the variability of response and even improvisation in the face of some problems, so to speak, which makes A scenario at this school would not necessarily be at another school.

Response: Thank you for your suggestions. I have tried to respond to each research question individually in the findings and discussion sections. In the findings section, there are three themes: Theme 1 - Perceptions of utilizing WSF, Theme 2 - Principal strategies, and Theme 3 - Teachers’ reactions. These three themes directly address Research Questions 1-3. By combining the findings across the three years, the dynamic changes in these three themes are shown. I then further address the research questions in the discussion section as I mentioned in the last paragraph of the data analysis. The first paragraph of the discussion addresses RQ1. The second and third paragraphs address RQ2. The fourth paragraph addresses RQ3. I also added one paragraph to conclude the findings and address RQ 1-3 at the same time.

  1. I am not sure if financial flexibility is adequate for discretionary application purposes as this could lead to misuse of funds on innovations that have not been tested to ensure successful application. For this reason, the initial plans, and the follow-up of these are relevant to be able to have clarity on the use of the funds. In addition, responding to emergent situations in a context of limited funds could affect those aspects that impact but are longer term.

Response: Thank you for your thoughtful feedback. I have clarified the meaning of flexibility in the policy implications, and have suggested ways to mitigate potential misuse of funds.

Round 2

Reviewer 2 Report

Dear Authors,

thank you for considering your time to improve the manuscript. Thank you for using the additional literature among your references. I accept your reasoning regarding the scope, even though I could still feel the need of a bit more thorough explication of the theoretical framework. I accept the manuscript in its current form. 

The level of English is high enough to accept it in its current form. 

Reviewer 3 Report

I think the researchers have addressed the suggestions. I have no further observations.